# A Fukui Analysis of an Arginine-Modified Carbon Surface for the Electrochemical Sensing of Dopamine

**DOI:** 10.3390/ma15186337

**Published:** 2022-09-13

**Authors:** Santhosh Kumar Revanappa, Isha Soni, Manjappa Siddalinganahalli, Gururaj Kudur Jayaprakash, Roberto Flores-Moreno, Chandrashekar Bananakere Nanjegowda

**Affiliations:** 1Department of Chemistry, University B.D.T. College of Engineering Visvesvaraya Technological University, Davangere 577004, India; 2Laboratory of Quantum Electrochemistry, School of Advacned Chemical Sciences, Shoolini University, Solan 173229, India; 3Department of Chemistry, Nitte Meenakshi Institute of Technology, Bangalore 560064, India; 4Departamento de Química, Centro Universitario de Ciencias Exactas e Ingenierías, Universidad Guadalajara, Blvd. Marcelino García Barragán 1421, Guadalajara C.P. 44430, Mexico; 5Department of Chemistry, Sri Adichunchanagiri First Grade College, Affiliated to University of Mysore, Hassan 573116, India

**Keywords:** sensors, DFT, graphite, voltammetry, interface

## Abstract

Amino acid-modified carbon interfaces have huge applications in developing electrochemical sensing applications. Earlier reports suggested that the amine group of amino acids acted as an oxidation center at the amino acid-modified electrode interface. It was interesting to locate the oxidation centers of amino acids in the presence of guanidine. In the present work, we modeled the arginine-modified carbon interface and utilized frontier molecular orbitals and analytical Fukui functions based on the first principle study computations to analyze arginine-modified CPE (AMCPE) at a molecular level. The frontier molecular orbital and analytical Fukui results suggest that the guanidine (oxidation) and carboxylic acid (reduction) groups of arginine act as additional electron transfer sites on the AMCPE surface. To support the theoretical observations, we prepared the arginine-modified CPE (AMCPE) for the cyclic voltammetric sensing of dopamine (DA). The AMCPE showed excellent performance in detecting DA in blood serum samples.

## 1. Introduction

Dopamine (DA) is a key neurotransmitter in animals and humans [1]. In 1958, Carlsson and his fellow researchers reported detailed research on the activities and functions of DA [1]. Carlsson was presented with the Nobel Prize in Physiology or Medicine in the year 2000. DA can activate and block the receptors on the postsynaptic membrane. Therefore, DA is considered an excitatory and inhibitory class of neurotransmitters. DA is a versatile neurotransmitter that functions in both classes and plays a critical role in the circulatory, renal, and hormonal systems. A high DA level is indicative of cardiotoxicity, which can result in irregular heart rhythms, high blood pressure, heart failure, and addiction to drugs. DA deficiency, on the other hand, has been linked to an increased risk of stress, Parkinson’s disease, schizophrenia, Alzheimer’s disease, and depression [2,3,4,5]. DA levels need to be measured, obviously, in order to have knowledge about their biological roles and the biological processes and mechanisms that are associated with them [6].

In the last ten years, a lot of research has gone into developing different ways to measure the amount of DA in body fluids, such as blood and urine. These methods include surface plasmon resonance [7], mass spectrometry, high-performance liquid chromatography [8], tandem mass spectrometry [9], and fluorescence [10]. Even though these highly reliable methods are generally well accepted, they have drawbacks of being expensive, time-consuming, and hard to do, and they need highly skilled people to do them [11].

The ideal biosensor should have attributes, such as a quick response time, simplicity of use, affordability, high sensitivity, and selectivity. By using voltage, the DA goes through a redox reaction and is converted to dopamine-o-quinone (DOQ). Due to its low cost, simple operation, quick response, high sensitivity, and viability of miniaturization, the electrochemical analytical technique for DA determination is an appealing method in this regard. By taking measurements using cyclic voltammetry (CV), the fabricated sensor’s sensitivity is kept track of. To perform the CV experiment, the choice of the working electrode is crucial [6,11].

Carbon paste electrodes (CPE) are some of the most commonly used working electrodes for voltammetric sensing applications. They have numerous advantages, such as easy surface renewability, low background currents, resistance to huge potential windows, and the possibility of miniaturization over other working electrodes for sensing applications [12,13]. The performance of bare CPE (BCPE), on the other hand, is highly dependent on its surface structure [14,15]. The electron transfer (ET) reaction between the CPE surface and analytes is critical for enhancing the detection limits of sensors and the electrode interface can be tailored to improve the electronic characteristics. Catalysts or modifiers affect BCPE’s surface characteristics (sensitivity and selectivity) for detecting analytes. Surfactant immobilization [14,15], electropolymerization from amino acids [6], and grinding modification [15] are some of the methods proposed by voltammetric researchers to prepare modified CPE (MCPE).

In recent years, arginine MCPE (AMCPE) has shown excellent applicability in electroanalysis, i.e., to sense molecules, such as cefotaxime [16], pyrogallol [17], fluoroquinolones [18], pyridoxine [19], riboflavin [19], carmine [20], DA [21], ascorbic acid [21], and uric acid [21]. Earlier experimental results prove that arginine will improve the reproducibility, stability, and repeatability of CPE. The sensitivity of the AMCPE is very high (to detect the redox-active molecules in real samples). Unfortunately, to date, there have been no reports that discuss how arginine improves the ET properties of the CPE interface at a molecular level. Therefore, there is a huge scope to understand the arginine-mediated ET process at the AMCPE interface. Numerous analytical approaches, such as X-ray diffraction, Raman spectroscopy, transition electron microscopy, high-resolution transition electron microscopy, atomic force microscopy, and scanning electron microscopy can be used to characterize the MCPE surface [22,23,24].

Quantum chemical modeling based on the density functional theory (DFT) can be applied to determine the interface features of MCPE. The use of DFT-based quantum chemical models to characterize the MCPE surface offers several advantages, including the ability to visualize the surface at the atomic level [6,14]. Modern analytical methods, such as high-resolution transition electron microscopy and scanning tunneling microscopy, can be used to visualize the carbon surface at the molecular level. However, these technologies are expensive and have limited availability in seeing the electrode surface. As a result, quantum chemical modeling is required to view the electrode surface on an atomic level. Aside from atomic-scale visualization, conceptual DFT-based quantum chemical models (Fukui functions [25,26,27] and dual descriptors [28,29,30]) will be useful in understanding electron transfer (ET) reactions and providing physical insights into modifier adsorption, local ET sites, and electrode surface energy levels. As a result, in the current research, we used a hypothetical conceptual DFT-based model to analyze the arginine-modified electrode interface for sensing applications.

## 2. Experimentation

### 2.1. Computational Methods

The Sinapsis tool [31] was used to draw the geometry of the arginine, graphene (GR), and arginine graphene (AGR) complexes; the same tool was also used to construct analytical Fukui plots and frontier molecular orbitals (FMO). Quantum chemical computations were performed using the deMon2k [32] program with VWN [33] correlation functional with DZVP [34] basis sets. As previously stated in the literature [35,36,37], auxiliary functions were built automatically.

### 2.2. Reagents and Chemicals

Sigma Aldrich (TM) provided arginine; Himedia provided DA, perchloric acid, graphite powder, and potassium chloride. Sodium dihydrogen phosphate and disodium hydrogen phosphate were utilized to make a 0.1 M phosphate buffer solution (PBS). Distilled water was used to make all of the compounds and reagents.

### 2.3. Instrumentation

A CHI-660 C (CH-Instruments, Inc., Bee Cave, TX 78738, USA) apparatus was used to conduct cyclic voltammetry studies. All of the studies were conducted in a standard three-electrode cell. A platinum wire served as a counter electrode, while saturated calomel (SCE) served as a reference electrode in the electrode system. Homemade bare CPE (BCPE) or arginine-modified CPE (AMCPE) were used as the working electrodes. All of the tests were conducted at room temperature [38,39,40,41,42,43].

### 2.4. Fabrication of AMPE

BCPE was kept in pH 6.5 PBS with 1 mM arginine solution to make LMCPE. Using CV, electropolymerization (deposition) was conducted by constantly applying potential between −0.2 and 1.2 V at a scan rate of 0.1 V/s for 10 cycles as shown in Figure 1. AMCPE was properly rinsed with distilled water after electrodeposition before electroanalysis, as described by our previous research article [19].

## 3. Results and Discussion

### 3.1. Computational Modeling of AGR Complexes

The arginine molecule sat on the graphite surface; N (rich sites) amine and (oxygen-rich sites) carboxylic acid groups present in the arginine molecule may affect the graphite surface’s redox electron transfer (ET) characteristics. ET sites on a graphite surface may be distinguished using quantum chemical calculations [14,44]. The BCPE was made up of carbon (graphite) particles with silicone oil (binder); the binder helps bind the graphite particles ET process mainly take place on the first layer of the graphite only. As a result, the graphene (GR) model was used as the BCPE surface. We previously modeled the BCPE surface using a 96-carbon atom graphene model [14,44] (and the current research used the same models) [14,44].

The arginine was deposited on the BCPE surface as a monomer, dimer, or polymer. Here, the single monomer was kept on the BCPE surface for modeling purposes (since our goal was to know the redox ET reactivity of arginine heteroatoms). The arginine was made up of amine, methylene, and the guanidine group. We placed arginine in four potential orientations on the BCPE surface to determine the optimum possible approach for arginine adsorption. Arginine can orient on the graphene surface by using the carboxylic acid group Figure 2a and the guanidine functional group Figure 2b or it can orient horizontally, as shown in Figure 2c,d. The arginine–GR complex generated by the GA interaction by the carboxylic acid group (Figure 2a) had the lowest energy (shown in Table 1). As a result, arginine interacted with the GR surface by its carboxylic acid group, as illustrated in Figure 2a, and it was chosen for future investigation. The interaction of arginine with the GR surface via the horizontal way (Figure 2c,d, and the guanidine group (Figure 2b) will not be investigated further.

### 3.2. FMO and Fukui Analysis of Arginine

The FMO hypothesis was suggested by K. Fukui et al. [45,46,47], and it offered a unique technique for anticipating ET reactivity utilizing the positions of the highest occupied molecular orbital (HOMO) and the lowest unoccupied molecular orbital (LUMO). During redox electrochemical processes, HOMO undergoes oxidation and LUMO undergoes reduction. The Fukui function is frequently used in electrochemistry to understand redox reaction pathways [6,14,44]. Simulations based on the analytical Fukui function have been used to analyze the electrochemical behavior of the CPE interface. The Fukui function may be defined using Equation Equation 1 [27].
(1)$$f^{\pm }\left(r\right)={\left(\frac{\partial \rho (r)}{\partial N}\right)}_{\nu (r)}^{\pm }$$

The electron density is denoted by ρ(r), N is the number of electrons in the system, and the + and − marks, respectively, signify electron addition and removal.

Figure 3a,c depict the HOMO and $f^{-}\left(r\right)$ arginine, respectively. The amine group of arginine contains HOMO orbitals and $f^{-}\left(r\right)$. Therefore, the guanidine group of poly(arginine) (in the presence of the amine group) MCPE appears to operate as oxidation centers at the electrode contact, based on these findings. Figure 3b,d depict the LUMO and $f^{+}\left(r\right)$ arginine, respectively. The carboxylic acid group of arginine contains LUMO orbitals and the reduction centers at the electrode interface.

### 3.3. FMO and Analytical Fukui Analysis of Arginine–GR

The reactive orbital space (ROS) FMO and analytical Fukui computational findings of arginine–GR are shown in Figure 4. The (ROS) HOMO of arginine–GR shown in Figure 4a and $f^{-}\left(r\right)$ shown in Figure 4c are located on edge sites of GR and heteroatoms of arginine (guanidine group). Therefore, terminal carbon atoms of GR and the guanidine group of arginine will help in the oxidation process of the AMCPE. The (ROS) LUMO of arginine–GR shown in Figure 4b and $f^{+}\left(r\right)$ Figure 4d are located on the terminal carbon atoms of GR and heteroatoms of the arginine (carboxylic acid group). Therefore, edge sites of GR and a carboxylic acid group of the arginine will reduce the process of the AMCPE. The results of the analytical Fukui are consistent with the FMO (pre-post) observations. As a result, the acquired results are more precise.

The radical properties of the modified carbon surface are also very important in electrochemistry. The average Fukui value ($f^{o}\left(r\right)$) can be used to predict regioselectivity in addition to free radicals. In this work, the average Fukui functions were calculated analytically [27].
(2)$$f^{o}\left(r\right)=\frac{f^{-}\left(r\right)+f^{+}\left(r\right)}{2}$$

The $f^{o}\left(r\right)$ of AMCPE is shown in Figure 5 and is $f^{o}\left(r\right)$ quite similar to $f^{-}\left(r\right)$.

### 3.4. Analytical Use of the AMCPE Interface for DA Sensing

#### 3.4.1. Analysis of DA at the AMCPE Interface

Chandrashekar et al. [21] previously studied the electrochemical behavior of DA at the AMCPE interface. In the current work, we analyze DA activity at the AMCPE interface to complement theoretical results. Figure 6 illustrates a CV of 10 μM DA in 0.1 M PBS (pH = 7.4) at a scan rate of 0.1 V/s using BCPE (purple line with dots) and AMCPE (green line). Currents are high at the AMCPE, possibly because of improved electrolyte movement and lower surface tensions at the electrode interface [11]. As shown in Figure 6 at the BCPE, ΔEp is 0.233 V, this shows that the redox electron transfer reactions between DA and BCPE are slow. However, at the AMCPE, ΔEp is 0.110 V, which shows that at AMCPE, the redox electron transfer is comparatively faster. Hence, arginine behaves similar to the catalyst for the electron transfer reactions between the DA and carbon surface.

#### 3.4.2. The Impact of Concentration of DA at AMCPE

Figure 7 represents a graph of the concentration of DA (5 to 20 μM DA) versus the anodic peak current. The obtained linear regression equation is Ipa = 0.3406c (concentration of DA in μM) + 13.44 with an R${}^{2}$ value of 0.9944.

We tested the AMCPE’s practical usefulness by determining DA in DA blood serum samples, as described in our previous article [11]. Table 2 displays the results. The recovery was satisfactory, indicating that the suggested procedures may be utilized to determine DA in pharmaceutical injections with a 95–96% recovery rate.

## 4. Conclusions

The results of the previous experiments and present voltammetric results reveal that arginine acts as a catalyst at the GR interface for enhancing electron transfer activity at the CPE interface. However, theoretical attempts to understand the arginine electron transfer action at the molecular level have not yet been explored. ET sites at the arginine–GR interface were observed in this study using the FMO (pre) and analytical Fukui functions (post) electron transfer Studies. The guanidine arginine group works as an extra electron donor site at the arginine–GR interface, while a carboxylic acid operates as a reduction center. As a result, conceptual DFT-based theoretical models will be extremely useful in understanding electrode insights and developing sensing applications.

## Figures and Tables

**Figure 1 materials-15-06337-f001:**
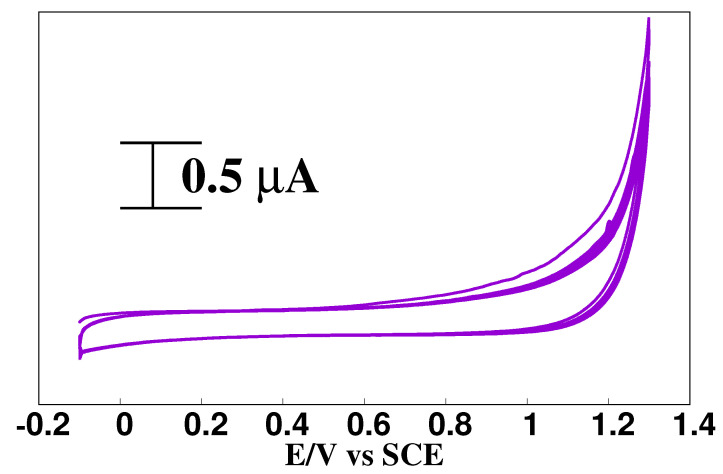
Electrochemical polymerization of l-arginine (1 mM) on CPE in PBS, pH 6.5, for ten multiple scans.

**Figure 2 materials-15-06337-f002:**
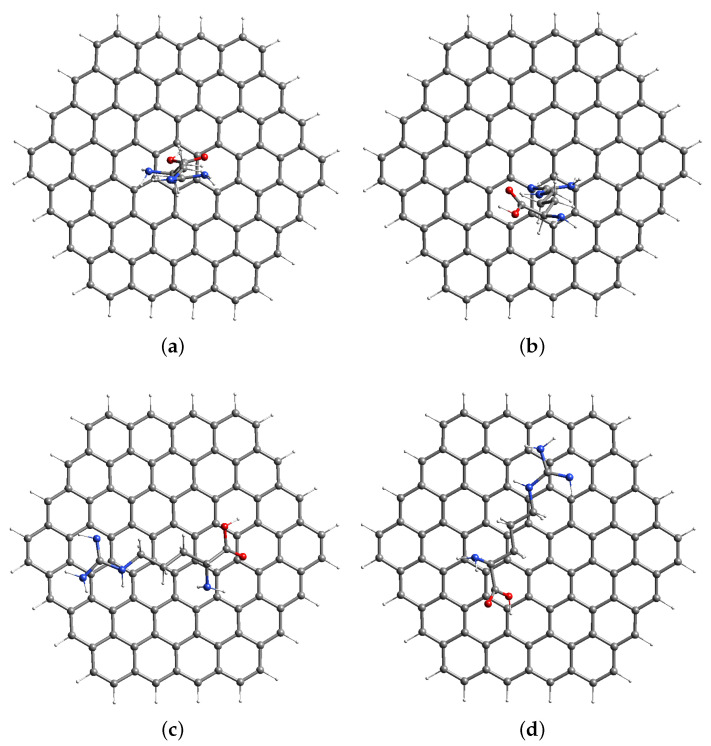
Selected folded conformers of arginine–GR. [H = white, C = grey, N = blue, and O = red]. (**a**) Arginine–GR a; GL interacting with GR with a carboxylic acid group. (**b**) Arginine–GR b; GL interacting with GR with an amine group. (**c**) Arginine–GR c; GL interaction with GR in a horizontal way. (**d**) Arginine–GR D; GL interaction with GR in a horizontal way.

**Figure 3 materials-15-06337-f003:**
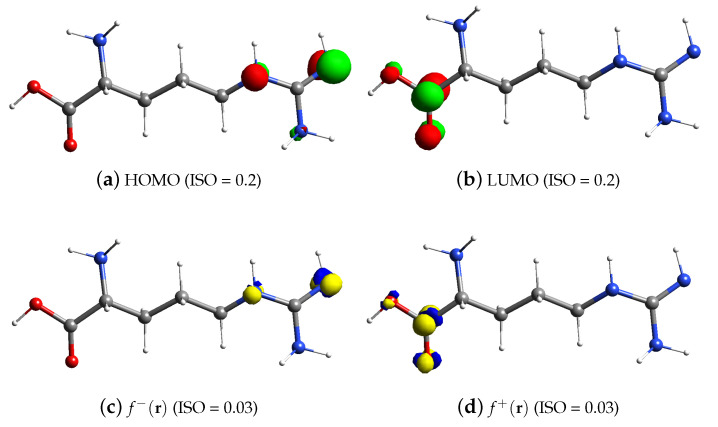
FMO and analytical Fukui analysis of arginine.

**Figure 4 materials-15-06337-f004:**
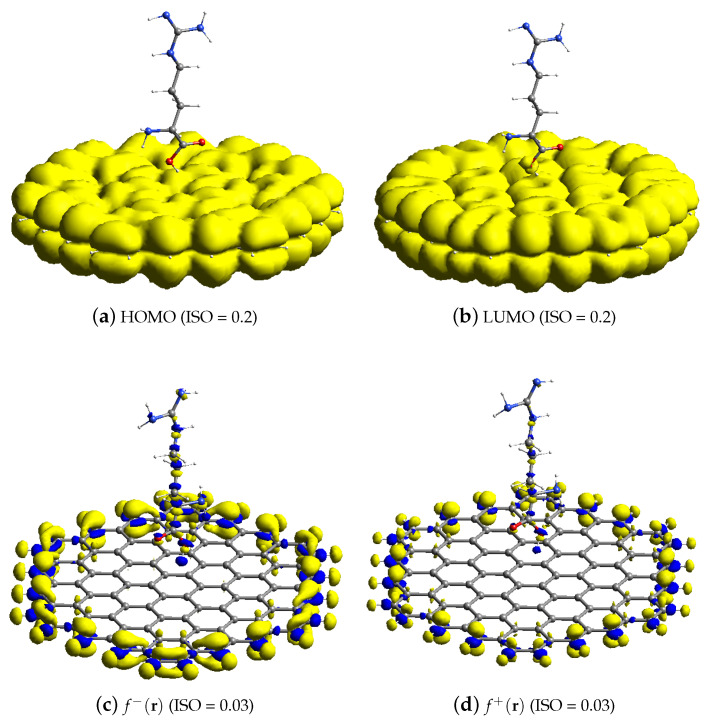
Average ROS FMO and Fukui analysis of arginine.

**Figure 5 materials-15-06337-f005:**
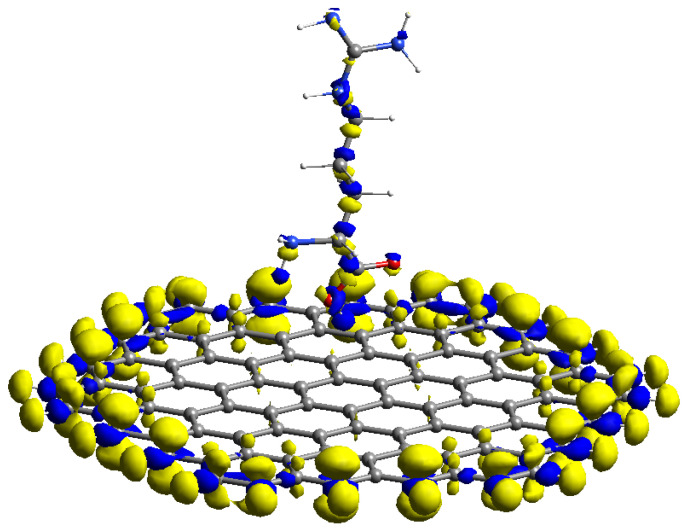
$f^{o}$(**r**) (ISO = 0.04).

**Figure 6 materials-15-06337-f006:**
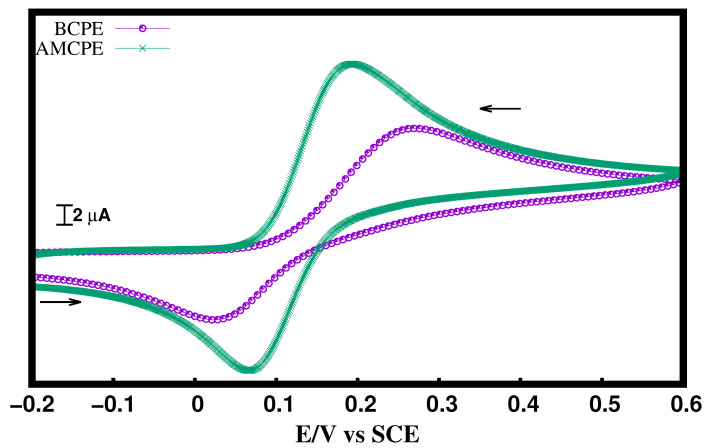
CV of 10 μM DA at BCPE and AMCPE.

**Figure 7 materials-15-06337-f007:**
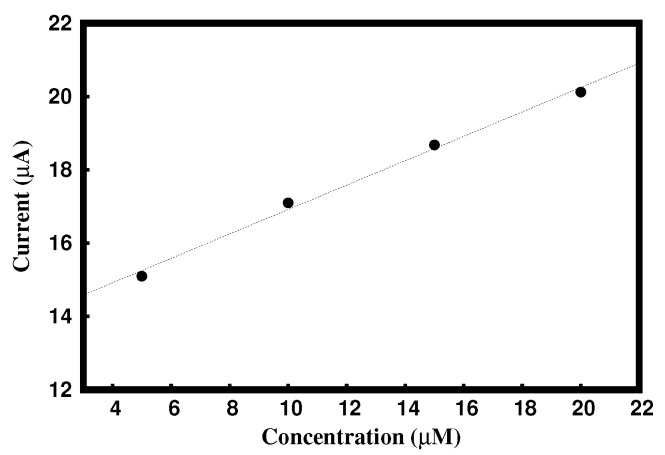
Graph of Ipa vs. concentrations.

**Table 1 materials-15-06337-t001:** Arginine–GR complexes energies as displayed in Figure 2.

Model	PBE/TZVP (kJ/mol)
Arginine–GR a	0.000
Arginine–GR b	9.466
Arginine–GR c	20.574
Arginine–GR d	8.135

**Table 2 materials-15-06337-t002:** Results of DA analysis in real samples.

SI/No.	DA Spiked	DA Sensed	Deviation	Recovery
	(μL)	(μL)	(μL)	(%)
1	5	4.75	−0.35	95.0
2	10	9.60	−0.40	96.0

## Data Availability

Not applicable.

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
