# Peer review of "A Fukui Analysis of an Arginine-Modified Carbon Surface for the Electrochemical Sensing of Dopamine"

_materials, 2022, doi:10.3390/ma15186337_

Round 1

Author Response

The article under discussion is the results of an experimental electrochemical study of a carbon electrode prepared from graphite paste subjected to treatment in an electric field and subsequent adsorption of arginine molecules on its surface. It has been established that the adsorbent enhances the voltammeric characteristics of the composite. To explain this fact, the paper presents the results of a quantum-chemical calculation of a model composite, which is a graphene domain in a necklace of hydrogen atoms and an arginine molecule adsorbed on it. The calculations were carried out using the standard program for four different arrangements of the molecule on the domain surface. The results of the calculation are analyzed from the point of view of the HOMO and LUMO and corresponding Fukui functions. The composite configuration corresponding to the maximum binding energy is established.

The latter is proposed as a model to explain the experimental results. In general, the article does not cause serious complaints: the problem statement, the results obtained and their analysis are described quite clearly, all the calculations are quite standard. Nevertheless, the legitimacy of using a simplified model of a virtual composite to describe the experimental behavior of a much more complex composite system remains insufficiently substantiated. The authors fully ignore radical essence of the graphene domain, which can and does drastically change configurations of both HOMO and LUMO as well as Fukui functions. The reviewer does not share the opinion of the authors that quantum chemical modeling, on the same level as experimental methods, is able to establish what happens in a real composite (line 79). In addition to the above, the greatest need for editing is required for the text on the lines 116-134. As a result of the foregoing, I consider it possible to publish an article in a journal after serious revising.

Dear reviewer, we are highly thankful for your careful analysis of our article. Based upon your suggestions we have improved our article. All the changes in the current modified version of the article are marked in red.

Here we are giving a detailed explanation and modification as per your comments.

  1. Nevertheless, the legitimacy of using a simplified model of a virtual composite to describe the experimental behavior of a much more complex composite system remains insufficiently substantiated.

Ans: We have selected the C96H24 model to mimic the graphene surface when compared to previous literature the current model contains much higher carbon atoms to make graphene models. In our previous article, we explained the suitability of models based on the length between C-C and Hirshfeld charges of each atom (The Journal of Physical Chemistry A 120 (45), 9101-9108).

We agree with the reviewer’s viewpoint, we have used graphite materials and the model is graphene. The electronic properties of graphene were much different from graphite. If we consider carbon paste electrodes, the silicon oil is unreactive and does not participate in electron transfer reactions. For modeling purposes, we are assuming that maximum electron transfer reactions take place at the first layer of graphite (graphene) therefore we are considering the graphene surface as the bare carbon paste electrode. The arginine deposits on the BCPE surface as a monomer, dimer, or polymer. Here single monomer is kept on the BCPE surface for modelling purposes (since our goal is to know the redox ET reactivity of arginine heteroatoms). The above-mentioned points are clear in the article from line number 110 to 120 in a crisp way.

  1. The authors fully ignore the radical essence of the graphene domain, which can and does drastically change configurations of both HOMO and LUMO as well as Fukui functions.

Ans: We agree with your point radical essence of the graphene domain is very important. As per your suggestion, we have added the radical Fukui plots, which indicate the radical susceptible sites of the modified carbon surface.

  1. In addition to the above, the greatest need for editing is required for the text lines 116-134. As a result of the foregoing, I consider it possible to publish an article in a journal after serious revising.

Ans: Thanks for pointing out the serious typo in the article, as per your suggestion now the typo between lines 116 to 134 is corrected. In the modified version the above-mentioned concerns are discussed between 122 to 126 lines.

Dear reviewer, we sincerely express our thanks to you for pointing out serious concerns about the article. Your comments are encouraging and helped us to improve the quality of the article.

Reviewer 2 Report

The manuscript by Santhosh et al. describes a combined theoretical and experimental effort to understand the role of arginine modification to the carbon surface in the detection of electrochemical sensing of dopamine. The claims are supported, and the conclusions are sound. My main concern regarding the theoretical portion of the manuscript is the relatively narrow selection of arginine-carbon interactions. For example, a) is graphene really a good representation of the surface CPE (graphite powder)? How would 3D carbon structures (inter-layer) affect the author’s conclusions? b) the arginine molecules are assumed as its linear form throughout the study, how would the conformational change in the arginine molecule affect the author’s conclusions?

Apart from this concern, here are a list of minor editorial improvements that the authors should consider –  

  1.  The authors should keep a consistency in their acronyms, e.g, both “dopamine” and “DA” appear multiple times throughout the manuscript; “ET” was defined multiple times; “LMCPE (Line 100, maybe the author meant AMCPE?)” was not defined throughout the manuscript 
  2. The title is unnecessarily lengthy, and therefore hard to grasp its gist – perhaps it could be modified as “a Fukui analysis of arginine modified carbon surface for the electrochemical sensing of dopamine”
  3. The authors should be more careful on the formatting – Table 1 lists four models as a, b, c, and ‘D’; ‘arginine’ was not capitalized in Line 122, to name a few. 

Author Response

Comments are attached 

Round 2

Reviewer 1 Report

The changes made by the authors significantly improved the article